# Phase II trial of neoadjuvant sitravatinib plus nivolumab in patients undergoing nephrectomy for locally advanced clear cell renal cell carcinoma

Jose A. Karam [1,2,6] ✉, Pavlos Msaouel [2,3,4,6] ✉, Cara L. Haymaker [2], Surena F. Matin [1], Matthew T. Campbell [3], Amado J. Zurita [3], Amishi Y. Shah [3], Ignacio I. Wistuba[2], Enrica Marmonti[2], Dzifa Y. Duose[2], Edwin R. Parra [2], Luisa Maren Solis Soto [2], Caddie Laberiano-Fernandez [2], Marisa Lozano[1], Alice Abraham[1], Max Hallin[5], Curtis D. Chin[5], Peter Olson, Hirak Der-Torossian[5], Xiaohong Yan[5], Nizar M. Tannir[3] & Christopher G. Wood[1,7]

Sitravatinib is an immunomodulatory tyrosine kinase inhibitor that can augment responses when combined with programmed death-1 inhibitors such as nivolumab. We report a single-arm, interventional, phase 2 study of neoadjuvant sitravatinib in combination with nivolumab in patients with locally advanced clear cell renal cell carcinoma (ccRCC) prior to curative nephrectomy (NCT03680521). The primary endpoint was objective response rate (ORR) prior to surgery with a null hypothesis ORR = 5% and the alternative hypothesis set at ORR = 30%. Secondary endpoints were safety; pharmacokinetics (PK) of sitravatinib; immune effects, including changes in programmed cell death−ligand 1 expression; time-to-surgery; and disease-free survival (DFS). Twenty patients were evaluable for safety and 17 for efficacy. The ORR was 11.8%, and 24-month DFS probability was 88·0% (95% CI 61.0 to 97.0). There were no grade 4/5 treatment-related adverse events. Sitravatinib PK did not change following the addition of nivolumab. Correlative blood and tissue analyses showed changes in the tumour microenvironment resulting in an immunologically active tumour by the time of surgery (median time-to-surgery: 50 days). The primary endpoint of this study was not met as short-term neoadjuvant sitravatinib and nivolumab did not substantially increase ORR.

Sitravatinib, an orally available tyrosine kinase inhibitor (TKI), targets key receptors involved in clear cell renal cell carcinoma (ccRCC) biology including the vascular endothelial growth factor receptor (VEGFR) family, c-MET, and the TAM (TYRO3, AXL, and MER) family. This not only inhibits angiogenesis but may mitigate immunosuppressive effects in the tumour microenvironment (TME) by reducing levels of myeloid-derived suppressor cells (MDSCs) and T regulatory (Treg) cells, and increasing the ratio of M1:M2-polarised macrophages[1,2]. Sitravatinib may therefore improve the efficacy of immune checkpoint inhibitors by producing a less immunosuppressive TME. Single-agent sitravatinib has shown activity in advanced ccRCC in a Phase 1/1b study in which patients had received a median of three prior treatment regimens[3]. Clinical activity and safety of sitravatinib plus nivolumab has also been demonstrated in

patients with advanced ccRCC whose disease had progressed on prior anti-angiogenic therapy[4].

Radical or partial nephrectomy is the current standard-of-care for locally advanced ccRCC[5]. However, while survival following nephrectomy is favourable, 20–40% of patients will develop metastases post-surgery[6]. Adjuvant immune checkpoint therapy can improve outcomes in patients with ccRCC compared with placebo indicating the potential value of immunotherapy approaches in the perioperative setting[7]. Although not currently approved for ccRCC, neoadjuvant therapy has been shown to downsize/downstage tumours ahead of surgery, and the safety and feasibility of this approach has been confirmed for several therapeutic agents, including TKIs such as sunitinib, sorafenib, pazopanib, and axitinib[8], and nivolumab immunotherapy[9]. While several combinations of immunotherapy with TKIs are available for metastatic RCC, there is currently a lack of data for the neoadjuvant setting. One Phase 2 study is ongoing (lenvatinib plus pembrolizumab [NCT04393350]), and a second (axitinib plus avelumab [NCT03341845]) recently reported partial responses in 30% of primary tumours following 12 weeks of treatment, confirming the validity of a neoadjuvant combination treatment approach[10–13].

We hypothesised that the combination of neoadjuvant sitravatinib with nivolumab would yield beneficial immunomodulatory responses in patients with ccRCC. This Phase 2 pilot study evaluated the efficacy and safety of neoadjuvant sitravatinib monotherapy lead-in, followed by the addition of nivolumab in patients undergoing nephrectomy for locally advanced ccRCC. The initial lead-in course of sitravatinib monotherapy facilitated exploratory correlative biomarker

analyses to enhance our understanding of the mechanisms of action of sitravatinib and nivolumab.

## Results

### Patients

From September 2018 to February 2020, 25 patients were enrolled. Of these, 20 patients received sitravatinib and were evaluable for safety (Fig. 1). The first patient was enrolled on October 9th, 2018 and the last patient was enrolled on February 10th, 2020. All 25 patients enrolled in the study underwent baseline biopsy. Of these, five discontinued the study (three had ineligible histology data, one was excluded for using concomitant treatment, which was not permitted by the study procedures and requirements). At mid-study, the 20 patients comprising the safety analysis population, all of whom had received sitravatinib treatment, were re-biopsied. All 17 of the patients in the efficacy-evaluable population underwent surgery, including one patient with bilateral disease who had two resections.

There was only one eligibility related protocol deviation in that an enrolled patient had systolic blood pressure greater than 150 mm Hg at screening. After four of the seven patients initially receiving sitravatinib 120 mg QD developed grade 3 hypertension, the starting dose was decreased to 80 mg QD for all remaining patients per the dose de-escalation mTPI plan; no further starting dose de-escalation was required.

Three patients in the safety analysis population were excluded from the efficacy analysis based on the retrospective determination of metastatic disease at baseline (for two patients) and study

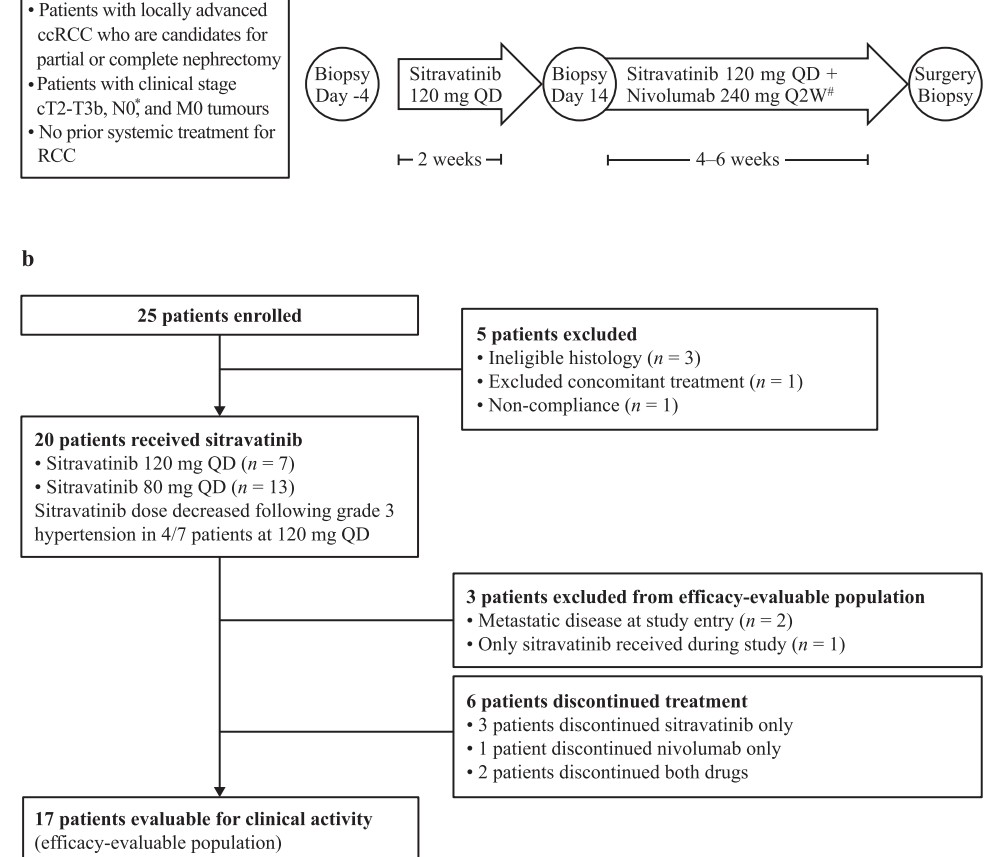

**Fig. 1 | Trial design and conduct. a** Study schema. **b** CONSORT diagram.
[*]Retroperitoneal lymph nodes ≤1 cm in size each considered N0. [#]Nivolumab 240 mg Q2W (Day 15, Day 29, and potentially Day 43); patients received nivolumab on Day 43 only if their surgery was expected to occur more than a week from that date; last dose of any drug was administered ≥72 h prior to surgery. ccRCC clear cell renal cell carcinoma, Q2W every 2 weeks, QD once daily, RCC renal cell carcinoma.

discontinuation after AE of Grade 3 lipase increased which was attributed to sitravatinib and occurred prior to first dose of nivolumab (for one patient). All 3 patients were enrolled per protocol based on information at screening and were therefore not considered protocol violations, but were also not included in the clinical activity evaluable population based on the protocol's prespecified statistical analysis plan. Thus, efficacy was evaluated in 17 eligible patients.

## Table 1 | Patient characteristics

| | Sitravatinib (120 mg) + nivolumab (n = 7) | Sitravatinib (80 mg) + nivolumab (n = 13) | Total (n = 20) |
|---|---|---|---|
| **Median age, years (range)** | 65.0 (55–72) | 61.0 (37–80) | 61.5 (37–80) |
| **Sex, n (%)** | | | |
| Male | 5 (71.4) | 11 (84.6) | 16 (80) |
| Female | 2 (28.6) | 2 (15.4) | 4 (20) |
| **Race, n (%)** | | | |
| Caucasian | 7 (100) | 12 (92.3) | 19 (95) |
| Other | 0 | 1 (7.7) | 1 (5) |
| **Ethnicity, n (%)** | | | |
| Hispanic/Latino | 1 (14.3) | 3 (23.1) | 4 (20.0) |
| Not Hispanic/Latino | 6 (85.7) | 9 (69.2) | 15 (75·0) |
| Not reported | 0 | 1 (7·7) | 1 (5.0) |
| **ECOG performance status, n (%)** | | | |
| 0 | 7 (100) | 12 (92.3) | 19 (95.0) |
| 1 | 0 | 1 (7.7) | 1 (5.0) |
| **Primary tumour stage, n (%)** | | | |
| T2b | 1 (14.3) | 0 | 1 (5.0) |
| T3 | 0 | 3 (23.1) | 3 (15.0) |
| T3a | 6 (85.7) | 10 (76.9) | 16 (80.0) |
| **Regional lymph node stage, n (%)** | | | |
| N0 | 7 (100) | 13 (100) | 20 (100) |
| **Distant metastasis stage, n (%)** | | | |
| M0 | 7 (100) | 11 (84.6) | 18 (90.0) |
| M1 | 0 | 2ᵃ (15.4) | 2ᵃ (10.0) |
| **Baseline hypertension, n (%)** | 5 (71.4) | 8 (61.5) | 13 (65.0) |

ECOG Eastern Cooperative Oncology Group.
ᵃTwo patients were presumed M0 at study entry, but later were found to have M1 disease after starting treatment.

In the safety-evaluable population, the median age was 61.5 years (range, 37–80 years); most patients had baseline hypertension (65%) and clinical T3 or higher-stage primary tumours (95%) (Table 1). Target lesions ranged from 16–124 mm at baseline (mean 76 mm).

### Radiological efficacy

As of December 1st 2021, the median extent of follow-up was 27.5 months (calculated from first dose date to last date known alive). The primary outcome was not met. In the efficacy-evaluable population, investigator-assessed confirmed objective response rate (ORR) was 11.8% (95% CI, 1.5–36.4; Exact test $p = 0.208$), comprising two radiologic partial responses (PRs), both in the sitravatinib 120 mg starting dose group (Table 2); an additional 15 patients (88.2%) had stable disease (Fig. 2a, b). No patient experienced either progressive disease prior to surgery or an increase in lesion size; median observed tumour shrinkage was 13.5% (range 0–33%). The secondary endpoint of median DFS was not reached at data cut-off; the estimated 12-month DFS probability was 94% (95% CI, 65–99); the estimated 24-month DFS probability was 88% (95% CI, 61–97) (Fig. 2c). The median follow up for DFS was 26 months (95% CI, 20.1 to 31.5 months); three patients in the efficacy-evaluable population had experienced a recurrence, after 11.8, 13.9, and 32.1 months (Fig. 2c).

### Safety

The cut-off date for the safety analysis (a key secondary endpoint) was July 2020 (median follow-up 9.4 months). Treatment-related AEs (TRAEs) of any grade occurred in all patients (100%) receiving sitravatinib plus nivolumab, with the majority being grade 1 or 2 (Table 3). Overall, grade 3 sitravatinib-related AEs occurred in nine patients (45%), grade 3 nivolumab-related AEs occurred in three patients (15.8%). The most common grade 3 TRAEs were hypertension (30%) and lipase increase (10%). There were no grade 4 TRAEs or grade 5 AEs; two patients (10.0%) experienced a total of five serious AEs (grade 4 urosepsis, grade 3 atrial fibrillation, and grade 2 urinary retention in one patient; grade 3 atypical pneumonia and grade 3 deep vein thrombosis [DVT] in one patient). There were no clinically important differences in the AEs observed between segment 1 (sitravatinib alone) and segment 2 (sitravatinib plus nivolumab).

Four of seven patients (57.1%) who received the starting 120 mg dose developed grade 3 hypertension, which led to dose reduction. Subsequent dose de-escalation qualifying events among the 13 patients who started treatment at 80 mg QD were grade 3 hypertension (two patients) and grade 3 DVT and pulmonary embolism (one patient each); this event rate did not require further dose de-escalation. All the dose de-escalation qualifying events of grade 3

## Table 2 | Efficacy outcomes (evaluable population)

| | Sitravatinib (120 mg) + nivolumab (n = 6) | Sitravatinib (80 mg) + nivolumab (n = 11) | Total (n = 17) |
|---|---|---|---|
| **Radiologic response following up to 8 weeks of treatment, n (%)** | 2 (33.3) | 0 | 2 (11.8) |
| 95% CIᵇ | 4.3–77.7 | 0.0–28.5 | 1.5–36.4 |
| P value | P = 0.033 | | P = 0.208 |
| **Radiologic response following up to 8 weeks of treatmentᵃ, n (%)** | | | |
| Partial response | 2 (33.3) | 0 | 2 (11.8) |
| Stable disease | 4 (66.7) | 11 (100) | 15 (88.2) |
| Progressive disease | 0 | 0 | 0 |
| **Disease recurrence, n (%)** | 1 (16.7) | 2 (18.2) | 3 (17.6) |
| **Median disease-free survival, monthsᶜ** | NE | NE | NE |
| 95% CI | 32.1 to NE | 13.9 to NE | 32.1 to NE |

Data cut-off date: 01 December 2021; median follow-up, 27.5 months.
CI confidence interval, NE not estimable, RECIST v1.1 Response Evaluation Criteria in Solid Tumors version 1.1.
ᵃBased on RECIST v1.1.
ᵇ95% CI calculated using the exact binomial method.
ᶜKaplan–Meier method.

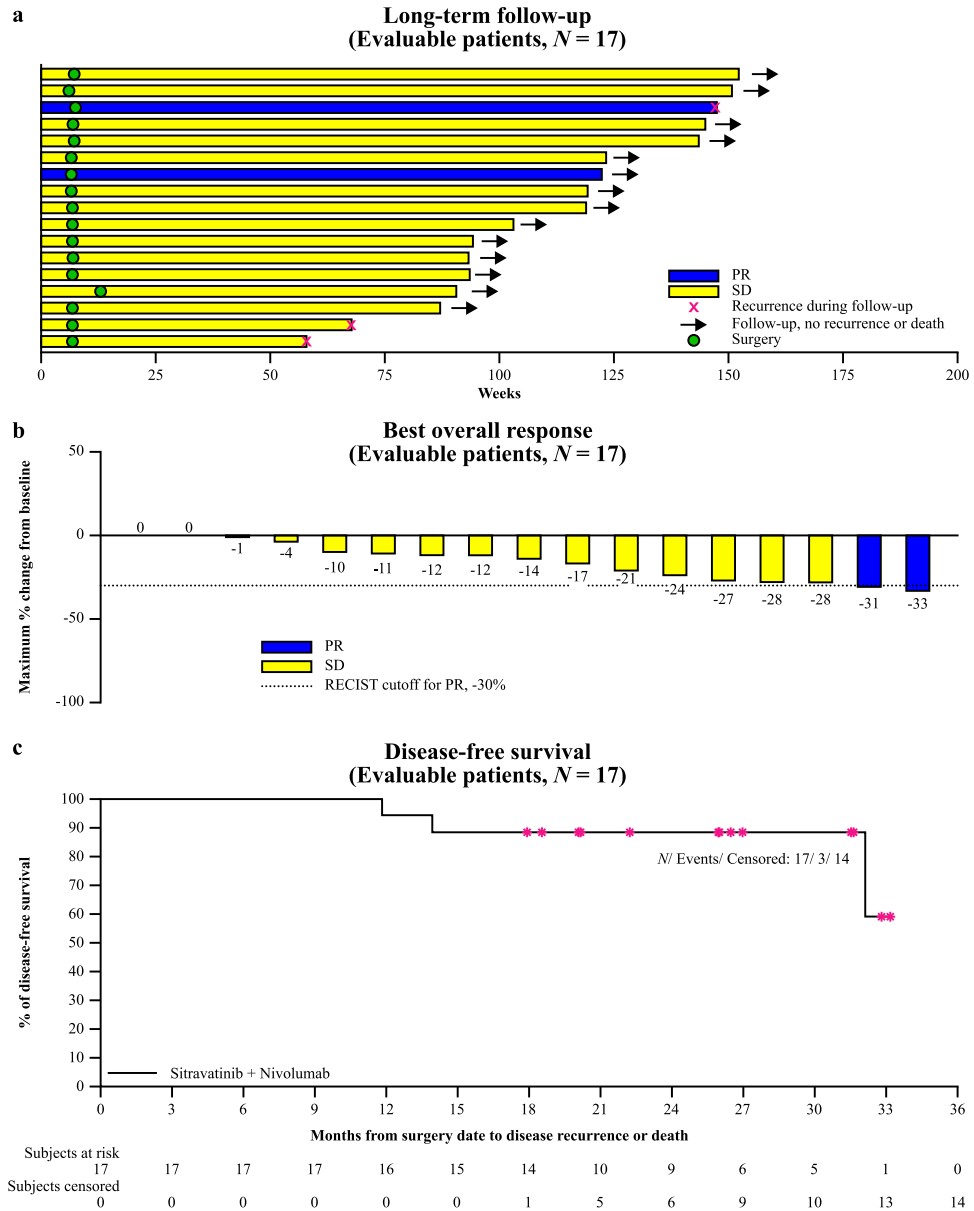

**Fig. 2 | Treatment response. a** Swimmer plot. **b** Waterfall plot. **c** Disease-free survival. *Censored. PR partial response, SD stable disease. Source data are provided in the Source Data file.

hypertension resolved with medication. Of the six patients overall with dose de-escalation qualifying hypertension events, four had pre-existing grade 1–2 hypertension.

TRAEs (related to either sitravatinib or nivolumab) leading to discontinuation occurred in six patients (30%); three patients discontinued sitravatinib (grade 3 increased lipase [Day 12]; grade 2 pancreatitis [Day 41]; grade 1 pancreatitis [Day 40]), one patient discontinued nivolumab (grade 2 pneumonitis [Day 30]), and two patients discontinued both sitravatinib and nivolumab (grade 3 pulmonary emboli [Day 18, sitravatinib-related]; grade 2 thyroiditis [Day 36, nivolumab-related]). The only TRAE leading to sitravatinib discontinuation for more than one patient was pancreatitis (grade 2 in one patient and grade 1 in one patient; both were asymptomatic and detected on laboratory investigation), both patients continued treatment with nivolumab.

At the sitravatinib doses of 80 mg and 120 mg, the respective median durations of sitravatinib treatment were 6.3 weeks (range, 2.6–7.3) and 7.1 weeks (range, 1.7–8.1); respective median numbers of nivolumab doses were 2 (range 1–2; n = 13) and 2·5

(range 2–3; n = 6). Nine patients (45.0%) had dose interruptions of sitravatinib.

Surgery was delayed in four patients (range of 3–38 days); one patient had a 38-day delay due to nivolumab-related thyroiditis that resolved, the remaining three patients had delays of 3–4 days. The secondary endpoint of median time from first dose of sitravatinib to surgery was 50 days (95% CI, 47–55). There were no complications during surgery. One patient experienced a Grade 3 complication using the Clavien-Dindo classification (temporary abdominal drain placement by interventional radiology for chyle leak 20 days after surgery)[14].

The 3 patients who were treated on protocol and included in the safety analysis population but not in the efficacy-evaluable population were not protocol violations. Per protocol's prespecified statistical analysis plan, the clinical activity evaluable population included patients who (1) have measurable disease (per RECIST 1.1) at baseline, (2) receive at least one dose of both sitravatinib and nivolumab, (3) have their on-study disease assessment prior to surgery, and (4) undergo surgery and are deemed disease-free (i.e. non-metastatic)

**Table 3 | Safety outcomes (any grade ≥10% in the total safety-evaluable population)**

| | Sitravatinib (120 mg) + nivolumab (n = 7) | | Sitravatinib (80 mg) + nivolumab (n = 13) | | Total (n = 20) | |
|---|---|---|---|---|---|---|
| | Any grade | Grade 3 | Any grade | Grade 3 | Any grade | Grade 3 |
| **Any TRAE, n (%)** | 7 (100) | 5 (71.4) | 13 (100) | 4 (30.8) | 20 (100) | 9 (45.0) |
| **TRAEsᵃ (≥10%), n (%)** | | | | | | |
| Hypertension | 5 (71.4) | 4 (57.1) | 7 (53.8) | 2 (15.4) | 12 (60.0) | 6 (30.0) |
| Dysphonia | 4 (57.1) | 0 | 6 (46.2) | 0 | 10 (50.0) | 0 |
| Oral dysesthesia | 3 (42.9) | 0 | 0 | 0 | 3 (15.0) | 0 |
| Lipase increased | 3 (42.9) | 1 (14.3) | 3 (23.1) | 1 (7.7) | 6 (30.0) | 2 (10.0) |
| Diarrhoea | 2 (28.6) | 0 | 6 (46.2) | 0 | 8 (40.0) | 0 |
| Amylase increased | 2 (28.6) | 0 | 1 (7.7) | 0 | 3 (15.0) | 0 |
| TSH increased | 2 (28.6) | 0 | 1 (7.7) | 0 | 3 (15.0) | 0 |
| Fatigue | 2 (28.6) | 0 | 7 (53.8) | 0 | 9 (45.0) | 0 |
| Myalgia | 2 (28.6) | 0 | 0 | 0 | 2 (10.0) | 0 |
| Hypothyroidism | 2 (28.6) | 0 | 0 | 0 | 2 (10.0) | 0 |
| ALT increased | 1 (14.3) | 0 | 5 (38.5) | 0 | 6 (30.0) | 0 |
| Constipation | 1 (14.3) | 0 | 2 (15.4) | 0 | 3 (15.0) | 0 |
| Decreased appetite | 1 (14.3) | 0 | 2 (15.4) | 0 | 3 (15.0) | 0 |
| Dizziness | 1 (14.3) | 0 | 2 (15.4) | 0 | 3 (15.0) | 0 |
| Headache | 1 (14.3) | 0 | 3 (23.1) | 0 | 4 (20.0) | 0 |
| Rash | 1 (14.3) | 0 | 2 (15.4) | 1 (7.7) | 3 (15.0) | 1 (5.0) |
| Pruritis | 0 | 0 | 3 (23.1) | 0 | 3 (15.0) | 0 |
| Glossodynia | 0 | 0 | 2 (15.4) | 0 | 2 (10.0) | 0 |
| Nausea | 0 | 0 | 2 (15.4) | 0 | 2 (10.0) | 0 |
| Pancreatitis | 0 | 0 | 2 (15.4) | 0 | 2 (10.0) | 0 |
| AST increased | 0 | 0 | 2 (15.4) | 0 | 2 (10.0) | 0 |
| Epistaxis | 1 (14.3) | 0 | 1 (7.7) | 0 | 2 (10.0) | 0 |
| Nail discolouration | 1 (14.3) | 0 | 1 (7.7) | 0 | 2 (10.0) | 0 |
| Hyperaesthesia | 0 | 0 | 2 (15.4) | 0 | 2 (10.0) | 0 |
| Thyroiditis | 0 | 0 | 2 (15.4) | 0 | 2 (10.0) | 0 |

*ALT* alanine transferase, *AST* aspartate aminotransferase, *TRAE* treatment-related adverse event, *TSH* thyroid stimulating hormone.
ᵃRelated to either sitravatinib or nivolumab.

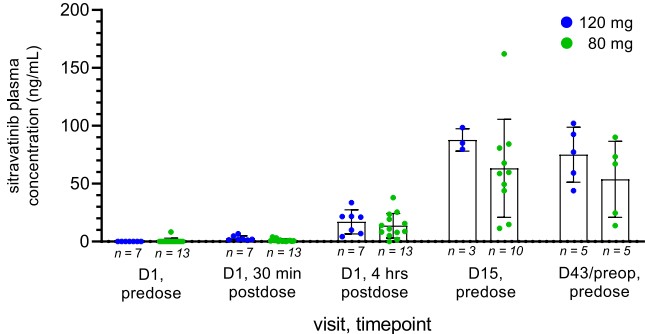

**Fig. 3 | Sitravatinib plasma concentrations.** At predose, 30 min, 4 h and 15 days after the first sitravatinib monotherapy dose, as well as 4 weeks following the addition of nivolumab (day 43). Reported are mean values with error bars indicating standard deviation. D day. Source data are provided in the Source Data file.

after surgery. Patients who discontinued treatment due to AEs or withdrawal of consent prior to their on-study disease assessment were not included in the clinical activity evaluable population. The presence of baseline metastatic disease for two of the patients was determined retrospectively after enrolling and initiating study treatment—the patients satisfied eligibility criteria per protocol based on information at time of screening. One patient had an adrenal mass thought to be an adrenal adenoma at baseline, but was subsequently determined to be a metastasis after surgery was done and pathology confirmed an adrenal metastasis and not an adrenal adenoma as initially thought at baseline, leading to study discontinuation. A second patient had lung metastases that were subtly apparent at baseline on retrospective review, leading to study discontinuation with no surgical resection. A third patient was not included in the efficacy analysis because of a Grade 3 lipase increased, considered to be related to sitravatinib (Table 3); the patient discontinued treatment before receiving nivolumab as per protocol, and did eventually have surgery but was not evaluable for clinical response due to the lack of a restaging scan prior to their operation. The guidance outlined in the protocol for Grade 3 or 4 sitravatinib-related non-hematological toxicities was to discontinue

sitravatinib, and patients who developed toxicities after receiving sitravatinib but prior to first nivolumab administration were considered for permanent discontinuation from study.

## Sitravatinib Pharmacokinetics
All patients were systemically exposed to sitravatinib following oral administration. At Day 15, the arithmetic mean pre-dose concentrations ($C_{trough}$) were 87.7 and 63.3 ng/mL at the 120 and 80 mg dose levels, respectively, and at Day 43 they were 75.0 and 53.7 ng/mL, respectively (Fig. 3). These secondary endpoint data suggest that sitravatinib reached steady state by Day 15, exposure increased in an approximately dose-proportional manner, and did not change following the addition of nivolumab.

## Tumoural expression of PD-L1 prior to therapy does not correlate with response
At baseline in the efficacy-evaluable population who had available tissue for staining, 1/14 tumours (7%) was classified as PD-L1 positive (≥1% PD-L1 expression). Per secondary endpoint analyses, there was no correlation between baseline PD-L1 expression and response; tumour size reductions were seen in PD-L1 positive and negative tumours (Fig. 4a). A subset analysis was conducted in samples where immunohistochemical (IHC) PD-L1 testing at both baseline and the time of surgery was feasible, as demonstrated in Fig. 4b. There was no association with tumour size reduction from baseline and increase in tumour PD-L1 expression following sitravatinib and nivolumab treatment (Fig. 4c; Supplementary Table S1).

## Proliferation and checkpoint receptor expression changes on T cells are restricted to the tumour
Amongst other secondary endpoints, multiplex IF analysis showed that sitravatinib monotherapy may be associated with an influx of CD3+ tumour-infiltrating lymphocytes (TILs) in some patients (Supplementary Fig. S1; Supplementary Table S2). Flow cytometry staining showed an influx of total immune cells (CD45+) into the tumour in some patients following sitravatinib monotherapy that was further increased following combination treatment (Fig. 5a). The CD4/CD8 ratio significantly increased following sitravatinib monotherapy compared with baseline (Fig. 5b–d). There was a significant decrease in natural killer (NK; CD45 + CD3-CD56dim) cells (Fig. 5e). Increased T-cell proliferation (Ki67) was observed in CD8+ TILs at the time of surgery in some patients (Fig. 5f). Baseline expression of checkpoint receptors, PD1, Tim3 and LAG3 on CD8+ and CD4 + TIL subsets showed that PD1 expression was highest in both TIL subsets (Supplementary Fig. S2A, S2B). Expression of Tim3 on either TIL subset was not impacted by therapy but both flow cytometry and multiplex IF showed that an increase in LAG3 + CD3+, LAG3 + CD8+, and LAG3 + CD4+ TILs was

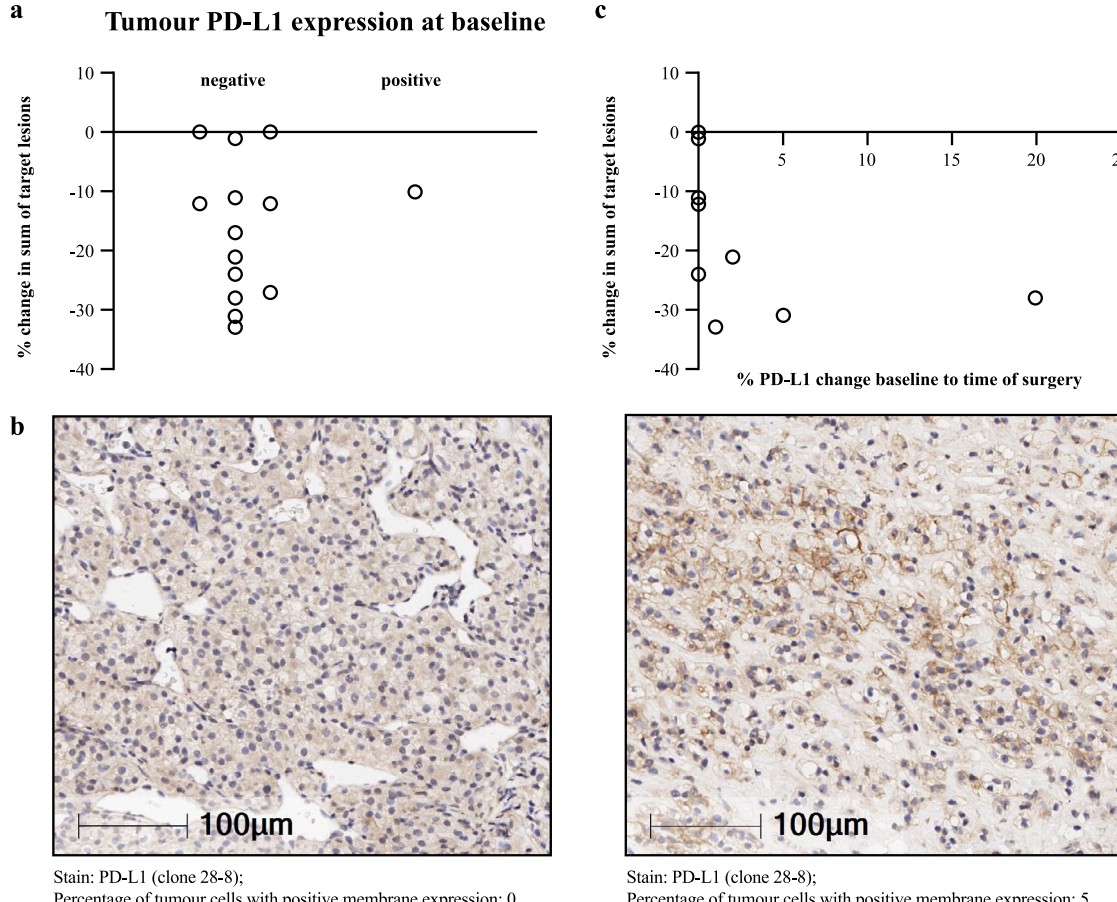

**Fig. 4 | Percentage change in tumour size in efficacy-evaluable patients. a** With negative versus positive PD-L1 tumour expression at baseline. **b** Example IHC images of PD-L1 expression at baseline (left) versus post-combination therapy at the time of surgery, at which time PD-L1 expression reached 5% (right) in subject 001, who had a 31% reduction in tumour size. **c** Percentage change in tumour size versus percentage change in tumour PD-L1 expression from baseline to time of surgery. IHC immunohistochemistry, PD-L1 programmed cell death ligand-1. Source data are provided in the Source Data file.

seen with sitravatinib plus nivolumab (Supplementary Fig. S2C–E). Expression of ICOS and OX40 was not found to be modulated over time on therapy on either CD8+ or CD4 + TIL subsets (Supplementary Fig. S3). Overall, upregulation of activation and inhibitory co-receptors as well as proliferation was found to be limited to the TIL population and was not observed in circulating peripheral blood T cells, suggesting that T-cell activation was restricted to the tumour (Supplementary Fig. S4). The TILs were subgated into CD4+ and CD8+ subsets and the CD4/CD8 ratio was generally <1 indicating a dominant CD8+ TIL population. CD4+ non-Treg TILs were assessed for co-expression of inhibitory receptors (CTLA-4, LAG3, PD-1, TIGIT and Tim3) as well as activation receptors (ICOS, OX40) and CD4+ T-effector cells were assessed for co-expression of LAG3, OX50, ICOS, and Tim3, data are shown in Supplementary Figs. S2 and S3. PD-1 expression could not be detected at post-nivolumab timepoints as the antibody clone used to stain by flow cytometry is blocked by nivolumab. In some cases, TIL activation and proliferation was detected at the time of surgery compared with baseline biopsies with an increase in LAG3 expression observed selectively in the CD8[+] TIL compartment.

In circulation, the T-cell fraction comprised a consistent frequency of ~50% with no change observed over time. The CD4/CD8 ratio was >1 with the CD4 subset being more abundant. Very low Treg frequencies were observed, and these did not change over time. Following the addition of nivolumab to the regimen, PD-1 expression on the T cell subsets could not be detected. For other checkpoint receptors assessed in the panel, TIGIT was found to be expressed on a subset of CD4+ T cells with some patients showing induction

post-sitravatinib. Tim3, LAG3, and ICOS were all weakly expressed (<1%) and expression was not changed over time (Supplementary Fig. S4). Expression patterns of Ki67 as a total or when stratified into low, moderate, or high expression did not change over time in circulating CD4 T cells, with most cells not proliferating. Overall, there was little impact observed in circulating T-cell activation, expression of checkpoint receptors, proliferation, or memory states.

## Macrophage tumour infiltration following sitravatinib monotherapy
Further secondary endpoint analyses showed that focusing on myeloid subsets revealed that sitravatinib monotherapy or in combination with nivolumab did not significantly change the number of macrophages, the percentage PD-L1 expression in CD68+ (Fig. 6), or the myeloid cell M1:M2 macrophage ratio (Supplementary Fig. S5) in tumour tissues. Higher median PD-L1 expression was observed on macrophages with combination therapy (18% of macrophages) versus sitravatinib monotherapy (3%) (Supplementary Fig. S5). Appreciable numbers of MDSCs were not observed in most patients and the presence of MDSCs was not associated with therapy (Supplementary Fig. S6).

## Gene expression profiling identifies pathways associated with immune activation, response, and resistance
The exploratory endpoint of gene expression profiling demonstrated significant downregulation of the angiogenesis-related receptor tyrosine kinases, *FLT1* (*VEGFR1*) and *KDR* (*VEGFR2*), and upregulation of the hypoxia Hallmark Gene Set Enrichment Analysis (GSEA) signature

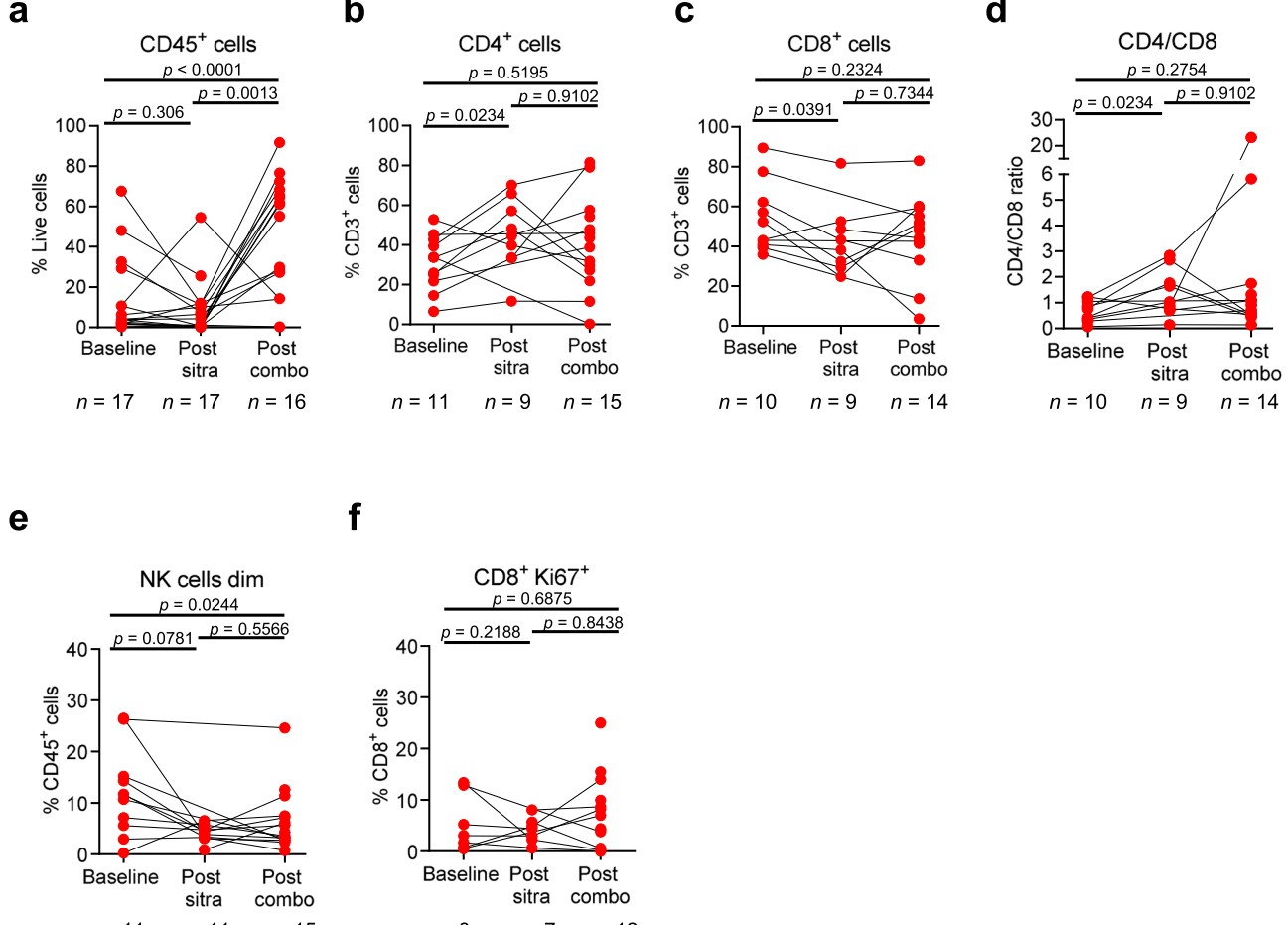

**Fig. 5 | Immune recruitment to the tumour tissue on therapy. a–f** Flow cytometry analyses of tumor tissue collected longitudinally. Total immune cell frequencies at baseline, post-sitravatinib, and at surgery post-combination therapy for live CD45+ cells (**a**), CD3 + CD4+ T cells (**b**), CD3 + CD8+ T cells (**c**), the CD4:CD8 T cell ratio (**d**), CD56 dim natural killer cells (**e**), and CD8 + Ki67+ T cells (**f**). Statistical significance between time points assessed was determined using a two-tailed, Wilcoxon matched pairs signed rank test and is indicated with a line at the top of the graph. The number of samples analysed per time point is shown for each graph. CD cluster of differentiation, NK natural killer. Source data are provided in the Source Data file.

## Discussion

In this study, 17 patients with locally advanced ccRCC received neoadjuvant treatment with sitravatinib plus nivolumab, and proceeded to nephrectomy with curative intent. Although the ORR of 11.8% did not meet the prespecified target of 30%, likely due to the short duration of neoadjuvant treatment, median DFS was not reached at data cut-off and the 24-month DFS probability was 88% (95% CI, 61–97). This result is comparable to the 24-month DFS probability of 77.3% (95% CI 72.8–81.1) observed in the KEYNOTE-564 phase 3 trial following a 12-month course of adjuvant pembrolizumab in patients with ccRCC[7]. Our study showed that patients with locally advanced ccRCC can safely proceed to surgery after a short period of combined neoadjuvant therapy and allowed investigation of the immunomodulatory effects of sitravatinib and nivolumab in the TME. Although limited by the small patient numbers, sample sizes, and its single-arm nature, the lead-in design of our study coupled with the detailed correlative biomarker analyses on single-agent and combination-treated tumour samples generated an informative data set.

In Phase 2 trials in both locally advanced and metastatic ccRCC, neoadjuvant therapy has been associated with reduced primary tumour burden prior to surgery[15]. Twelve weeks' neoadjuvant axitinib (VEGFR TKI) was clinically active and reasonably well-tolerated in patients with locally advanced ccRCC (median 28% reduction in primary renal tumour diameter; PR 46%)[16]. Similarly, 8 weeks' neoadjuvant sunitinib (VEGFR and platelet-derived growth factor receptor [PDGFR] TKI) was safe and feasible in patients with locally advanced or metastatic ccRCC (median 21% reduction in primary renal tumour diameter; PR 29%)[17]. In patients with metastatic ccRCC, 12–14 weeks' neoadjuvant pazopanib (VEGFR and PDGFR TKI) provided clinical benefit to 84% of patients, with a median 14% reduction in primary tumour size[18]. Perioperative nivolumab showed preliminary feasibility and safety with no surgical delays or complications in a Phase 2 study in non-metastatic high-risk ccRCC[9]; a Phase 3 randomised study comparing perioperative nivolumab to observation in patients with localised RCC undergoing nephrectomy is underway[19]. In the only other study of combination treatment reported to date, 12 weeks of neoadjuvant axitinib plus avelumab (anti-PD-L1 checkpoint inhibitor) in high-risk non-metastatic ccRCC produced partial responses in the primary tumours of 30% of patients, which correlated with longer-term outcomes (92% of responding patients were disease-free after a median follow-up of 23.5 months)[13]. Together with our own data, these results confirm that combination neoadjuvant therapy offers potential clinical benefit to patients scheduled for surgery.

Results from correlative assessments support modulation of the tumour microenvironment by sitravatinib via hypoxia and

after sitravatinib monotherapy and sitravatinib plus nivolumab. In addition, the interferon (*IFN*)-γ response gene signature was increased when sitravatinib was combined with nivolumab compared with baseline (Fig. 7a–c).

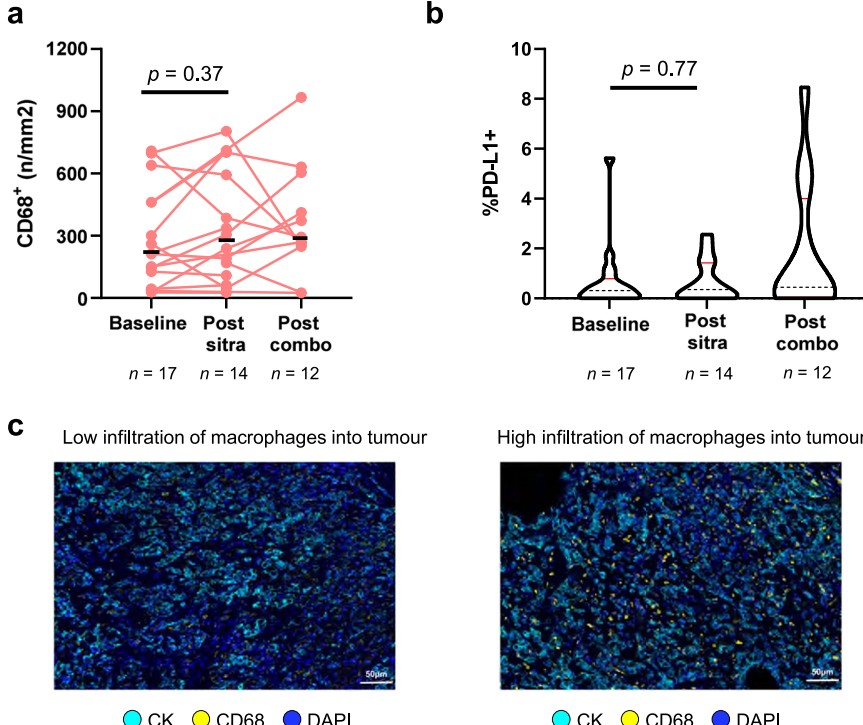

**Fig. 6 | Macrophage infiltration and expression of PD-L1 is not modulated on-treatment. a** Multiplex IF results showing CD68+ macrophage densities at baseline, post-sitravatinib monotherapy, and at surgery post-combination therapy. The median density is indicated with the black line. **b** Percentage PD-L1 expression by CD68+ myeloid cells at baseline, post-sitravatinib monotherapy, and post-combination therapy. The median is indicated with the dashed line in the violin plot. **c** Example images of cases with low and high macrophage infiltration into tumour. A paired, two-tailed Student's *t* test was used to assess changes from baseline to post-sitravatinib. The number of samples analysed per time point is shown for each graph. Abbreviations: CD cluster of differentiation, CK cytokeratin, DAPI 4, 6-diamidino-2-phenylindole, IF immunofluorescence, PD-L1 programmed cell death ligand-1. Source data are provided in the Source Data file.

immunomodulation pathways; this modulation was further enriched with the combination of sitravatinib and nivolumab. Gene expression analyses showed that the angiogenesis biomarkers *KDR* (*VEGFR2*) and *FLT1* (*VEGFR1*) were two of the top genes downregulated following sitravatinib monotherapy, indicating potent inhibition of angiogenesis, and confirming a key mechanism of action of sitravatinib. The GSEA analysis showed significant upregulation of the hypoxia gene signature with sitravatinib and further enrichment with the combination of sitravatinib and nivolumab. In addition, several immune-related GSEA pathway signatures were increased, providing compelling evidence that sitravatinib stimulates the immune system. The decrease in *TGF-β*, *E2F* targets, and *G2M* checkpoint signatures following sitravatinib treatment further demonstrates reversal of the immune suppressive TME and blockade of cell proliferation by sitravatinib.

Increased CD8+ T cell infiltration and correspondingly low CD4/CD8 T cell ratio are associated with poor prognosis in ccRCC[20–22]. In our multiplex IF analysis, sitravatinib monotherapy increased the CD4/CD8 ratio in the ccRCC tumors. Sitravatinib also appeared to increase macrophage density in the tumor, but not increase the M1:M2 macrophage ratio. This finding contrasted with previous nonclinical and clinical studies (e.g. resectable oral cavity squamous cell carcinoma) demonstrating sitravatinib increasing M1:M2 macrophage ratio possibly due to TAM receptor inhibition[1,2,4], supporting modulation or removal of key immunosuppressive cell types (M2 macrophages) in the TME by sitravatinib. Increased PD-L1 expression on macrophages is likely an effect of nivolumab. Most tumours (16/17) were PD-L1 negative at baseline, which is lower than expected since other studies have reported PD-L1-positive rates (i.e., expression >0%) of around 30%[23]. However, we did observe some tumour responses in this largely PD-L1-negative population, consistent with literature showing that patients whose disease is PD-L1 negative by IHC can still achieve clinical benefit with anti-PD-1/anti-PD-L1 therapies[24].

GSEA analysis showed that the *IFN-γ* response pathway was increased with sitravatinib monotherapy and further increased with sitravatinib plus nivolumab. The increased IFN-γ response gene signature noted in our study following treatment with sitravatinib plus nivolumab may sensitize tumors to anti-CTLA-4 immune checkpoint inhibition[25]. We have accordingly activated a phase 1 trial (NCT04518046 at clinicaltrials.gov) to determine the safety and efficacy of adding the CTLA-4 inhibitor ipilimumab to sitravatinib plus nivolumab in patients with advanced ccRCC[26]. Furthermore, changes in the TME were concordant between multiplex IF and flow cytometry analyses showing an increase in LAG3-expressing TILs over time in more responsive patients. This finding suggests the presence of suppressive pathways that can be induced by the immune activation observed in this study and which may be actionable targets for future combination treatments. In addition, multiplex IF demonstrated an increased infiltration of LAG3 + CD3+ TILs during treatment compared with baseline, suggesting that combination therapy induced specific TME changes resulting in an immunologically active state before surgery. In primary ccRCC, LAG-3 expression is associated with shorter survival, may be an indicator of poor prognosis[27], and could be one reason for lack of response in some of our patients.

In conclusion, this Phase 2 study demonstrated that short-term neoadjuvant sitravatinib plus nivolumab does not substantially increase ORR but may modulate the immune microenvironment and yield high DFS probabilities in locally advanced ccRCC. Despite the presence of qualifying toxicities leading to dose de-escalation after the first 7 patients enrolled, all patients regardless of starting dose of sitravatinib were able to safely proceed to surgery. Biomarker findings were consistent with the mechanism of action of sitravatinib, demonstrating that angiogenesis inhibition and anti-tumoural immunomodulation were the key pathways that were upregulated with sitravatinib and further augmented with combination treatment. These data

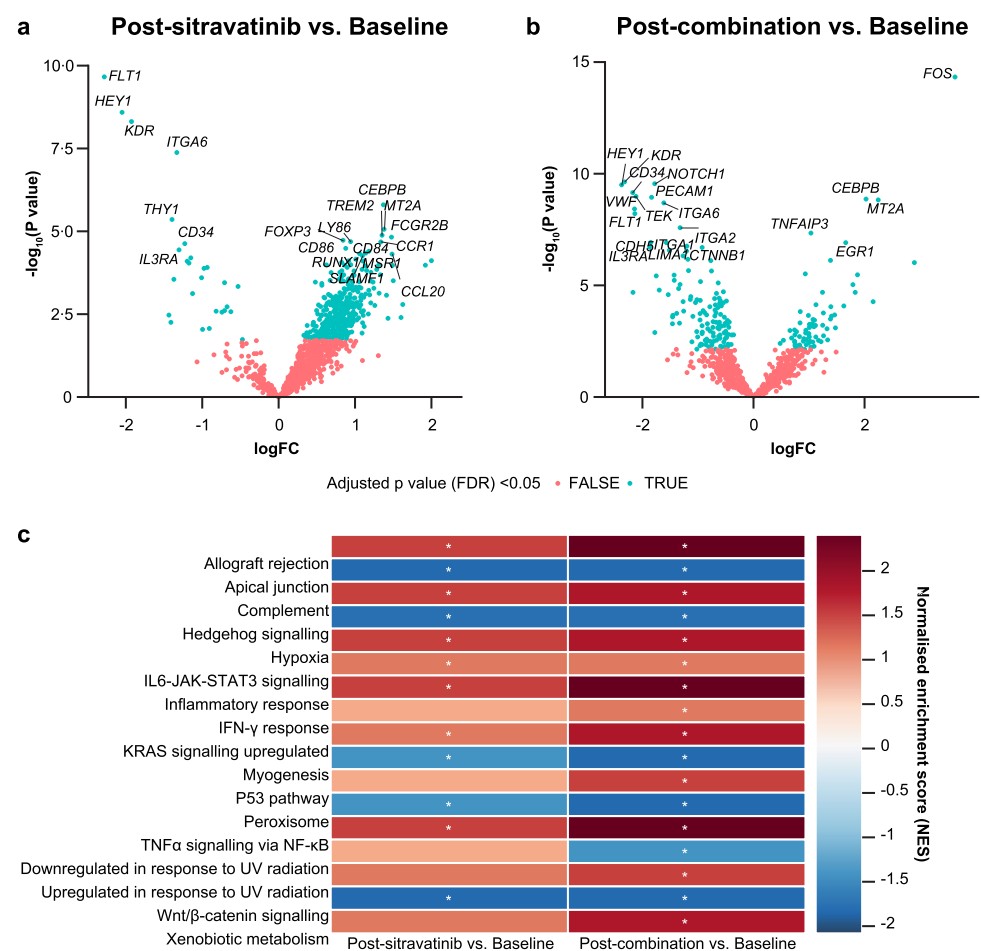

**Fig. 7 | Gene expression profiling. a** Top angiogenesis-associated genes down-regulated post-sitravatinib versus baseline. **b** Top genes differentially expressed post-combination therapy versus baseline. Differential expression analysis was performed using limma-trend. FDR values are adjusted *p*-values for multiple comparisons using the Benjamini & Hochberg method. **c** Top Hallmark Gene Set Enrichment Analysis (GSEA) pathways regulated post-sitravatinib and post-combination therapy versus baseline including immune (allograft rejection, complement, inflammatory response) and tumour (*E2F, G2M*) pathways regardless of tumour response. GSEA was used to determine the NES (normalized enrichment score) and FDR. FDR is adjusted for both gene set size and multiple hypotheses testing. Precision Immuno-Oncology panel on the HTG EdgeSeq platform, analysed using Limma V. 3.40.9. *FDR < 0.25. FDR false discovery rate, IFN interferon, logFC log fold change, NES normalised enrichment score, TNF tumour necrosis factor, UV ultraviolet. Source data are provided in the Source Data file.

demonstrate an immune response with sitravatinib that is supportive of its action in favourably modulating the TME.

## Methods

### Trial oversight
This study was approved by the institutional review board/independent ethics committee of The University of Texas MD Anderson Cancer Center and conducted in accordance with Good Clinical Practice guidelines, defined by the International Conference on Harmonisation. All patients provided written informed consent to participate based on the principles of the Declaration of Helsinki.

### Patients
Eligible patients were aged ≥18 years with previously untreated locally advanced ccRCC without evidence of metastatic disease. All patients underwent an initial diagnostic biopsy of their renal lesion to confirm clear cell histology. Eligible patients had clinical stage cT2-T3b, N0, M0 tumours, with retroperitoneal lymph nodes ≤1 cm in size (considered clinical N0) and were candidates for partial or radical nephrectomy. Additional key inclusion criteria were measurable disease according to Response Evaluation Criteria in Solid Tumors version 1.1 and an Eastern Cooperative Oncology Group performance status score of 0 or 1. Key exclusion criteria included inability to undergo a baseline tumour

biopsy; a clinical status indicating the need for immediate (within 6 weeks) surgery, regardless of whether neoadjuvant therapy was to be administered; autoimmune disease; or any current/prior use of an immunosuppressant (>10 mg daily prednisone equivalent). The full trial protocol is available in the supplementary note.

### Study design
This Phase 2, open-label, single-arm study (516-002 trial; clinicaltrials.gov identifier: NCT03680521) included two sequential preoperative treatment segments (Fig. 1a). In the first segment, sitravatinib monotherapy was administered for 2 weeks; in the second segment, combination sitravatinib plus nivolumab treatment was administered for ≥4 weeks (maximum 6 weeks, allowing surgical scheduling flexibility). After completing neoadjuvant therapy, patients underwent a pre-surgical restaging scan, followed by planned resection, either partial or radical nephrectomy. Patients were subject to a 48-h preoperative hold of all study drugs prior to surgery. No study drug was administered post-surgery.

Sitravatinib was administered orally once daily (QD) at the starting dose of 120 mg. The modified Toxicity Probability Interval (mTPI) method was used to set rules on a dose de-escalation plan to monitor and limit toxicity of the starting dose of sitravatinib in the combination regimen in the neoadjuvant setting[28]. Nivolumab was administered at

the recommended dose, 240 mg every 2 weeks as a 60-minute intravenous infusion. Dose delays and modifications for adverse events (AEs) were permitted for sitravatinib.

### Endpoints and assessments

The primary study objective was to evaluate clinical activity of the combination regimen; the primary endpoint was objective response rate (ORR) defined as the percentage of patients achieving a radiographic complete response (CR) or partial response (PR), per RECIST v1.1, prior to surgery. With currently available treatments, the percentage of patients with a point in time ORR prior to surgery was assumed to be 5% (null hypothesis) and this rate was thus considered uninteresting. The target percentage of patients with a point in time objective response prior to surgery using sitravatinib and nivolumab was assumed to be 30% (alternative hypothesis). Secondary endpoints were safety; pharmacokinetics (PK) of sitravatinib; immune effects, including changes in programmed cell death–ligand 1 (PD-L1) expression; time-to-surgery; and disease-free survival (DFS). Exploratory endpoints included biomarker analyses of the effect of sitravatinib alone and in combination with nivolumab.

Imaging was used for disease assessments, with an allowable window of 4 weeks prior to first study treatment for screening/baseline evaluation and within 1 week of planned surgery for on-study disease assessment. Per protocol, patients would be followed for survival every 6 months from the last study visit for at least 3 years or until death, disease recurrence, or loss to follow-up; disease recurrence was based on off-study imaging assessments performed per standard-of-care for post-nephrectomy patients. AEs were graded using the National Cancer Institute Common Terminology Criteria for Adverse Events version 5.0. Data were collected using the Medidata RAVE platform version 2017.2.2.

### Assessments

Baseline disease assessments were performed using computed tomography (CT), X-ray (radiography), or magnetic resonance imaging (MRI). Subsequent, on-study disease assessment included imaging of all known and suspected sites of disease identified in the preoperative setting (ie, CT or X-ray of the chest, CT, or MRI of the abdomen, and, if clinically indicated, whole body bone scan and CT with contrast or MRI of the brain and evaluation of any superficial lesions). The allowable window for imaging was 4 weeks prior to first study treatment for screening/baseline evaluation and within 1 week of planned surgery for on-study disease assessment. Imaging results were evaluated by the investigator to assess disease response per RECIST v1.1. Blood samples for PK evaluation were collected at specified timepoints prior to and following study treatment dosing. Safety assessments were conducted at the initiation of study treatment and at each clinic visit. Per protocol, patients would be followed for survival every 6 months from the last study visit for up to 3 years or more until death, disease recurrence, or loss to follow-up; disease recurrence will be based on imaging assessments performed off study per standard of care for patients post-nephrectomy.

Dose-limiting toxicities (DLTs) were assessed to guide adjustment of the starting dose of sitravatinib after 3 weeks for the first six treated patients or earlier if ≥2 patients were suspected of experiencing DLTs. The dose of sitravatinib was to be decreased if >2 of the first six patients experienced DLTs. DLTs were defined as non-haematological grade 4 AEs; non-haematological grade 3 AEs except (a) manageable nausea, vomiting, and diarrhoea persisting <2 h, (b) uncomplicated electrolyte abnormalities resolved within 72 h, (c) fatigue persisting <8 days, and d) amylase or lipase elevation not associated with pancreatitis; any toxicity that delayed surgery by >2 weeks.

Biomarker expression analyses were conducted on core needle biopsies taken at baseline (timepoint 1), during sitravatinib monotherapy at day 14 (timepoint 2), and on resected specimens at the time of surgery during sitravatinib plus nivolumab (timepoint 3). Blood samples for correlative studies were collected at screening, day 1, day 15, day 29, and day 43/surgery. Biomarker analyses included immunohistochemistry (IHC) for PD-L1 expression (using clone 28-8; cat# ab205921 [Abcam, Cambridge, UK] as used in the Agilent PharmDx system), multiplex immunofluorescence (IF) profiling, gene expression profiling using NanoString nCounter and HTG EdgeSeq technologies, and tissue and blood flow cytometry.

### Discontinuation criteria

Patients were permitted to discontinue from study treatment or from the study at any time at their own request, or by the discretion of the Investigator or Sponsor for safety, behavioural reasons, or for significant protocol violations. Further discontinuation reasons included objective disease progression, global deterioration of health, adverse events (as detailed below), loss to follow-up, refusal for further treatment, study termination by Sponsor, and death.

Permanent discontinuation of sitravatinib was implemented in the following circumstances:

- Grade 3 or 4 febrile neutropenia
- Grade 4 thrombocytopenia of any duration
- Grade 4 hypertension
- Grade ≥3 palmar-plantar erythrodysaesthesia
- Grade ≥3 haemorrhage
- Grade ≥2 thrombotic events (including thrombosis, pulmonary embolism, myocardial infarction, cerebrovascular accident, and thromboembolic event)
- Development of nephrotic syndrome
- Grade ≥3 increased transaminase
- Immune-mediated hepatitis
- Use of any radiation or earlier-than-planned surgery to manage cancer lesions
- Pregnancy
- Immune-mediated colitis
- Increase in aspartate aminotransferase and/or alanine aminotransferase ≥3 × the upper limit of normal (ULN) and bilirubin ≥2 × ULN but without concurrent increases in alkaline phosphatase, that is not attributable to liver metastases or biliary obstruction.

Permanent discontinuation from the study was considered in the following circumstances:

- If treatment with sitravatinib was withheld for ≥14 consecutive days
- If after receiving sitravatinib, but prior to any nivolumab dosing, patients developed toxicities that prevented the first nivolumab administration
- If significant hypertension recurred (this could also be addressed through medical management and/or dose reduction)
- In the event of treatment-related, grade ≥2 decreased ejection fraction
- For patients requiring acute hospitalisation for treatment of congestive heart failure.

Non-haematological toxicities of grade ≥3 and considered to be sitravatinib-related were managed with permanent discontinuation of sitravatinib. With the occurrence of grade 3 toxicities of nausea, vomiting, diarrhoea, and laboratory abnormalities that were adequately managed by routine supportive care (such as anti-emetics, anti-diarrhoeals, or electrolyte supplementation) and persisted for ≤72 hours, grade 3 fatigue lasting ≤8 days, or grade 3 amylase or lipase elevation, then sitravatinib treatment may be interrupted until resolution of toxicity to grade ≤1 or to baseline value and subsequently resumed at the same dose. Treatment with sitravatinib was discontinued in the presence of ≥2 g of proteinuria/24 h but could be restarted when protein levels decreased to <2 g/24 h.

Required dose modifications (i.e., interruption, dose reduction, or discontinuation) for nivolumab were performed per the current

OPDIVO® US Prescribing Information (USPI OPDIVO [nivolumab])[29], in addition to potential dose modifications for sitravatinib. Furthermore, patients permanently discontinued nivolumab in the presence of any grade 3 or 4 immune-related AEs; whereas sitravatinib could be resumed at the same or lower dose at the discretion of the Investigator until the event stabilised to grade ≤1.

### Tissue-based assays

**HTG EdgeSeq.** The HTG EdgeSeq gene expression platform is a high throughput next-generation sequencing (NGS)-based assay that utilises low sample input and a unique nuclease protection chemistry to simultaneously assess gene expression levels in multiple genes. We used the 1392 gene HTG EdgeSeq Precision ImmunoOncology panel (PIP) to assess tumour immune response from a total of 37 patient samples.

Three batches of samples were run using the PIP; batch 1 samples comprised 16 formalin-fixed paraffin embedded (FFPE) samples with two 5 µm core needle biopsy sections, while batches 2 and 3 comprised 23 samples with three 5 µm core needle biopsy sections. Of the 37 samples, only two samples had a total surface area <6 mm$^2$ (minimum required input into assay) and thus failed the sample input criterion.

All samples meeting the minimum input area were processed according to the manufacturer's protocol. Briefly, the tissue was removed from the slides with a scalpel, lysed using proteinase K and HTG's denaturation oil, and underwent target protection and clean up on the HTG EdgeSeq processor. Following this process, sequencing adaptors and barcodes were added and the library amplified. Amplified libraries were quantified with the Kapa library quantification kit, pooled, and sequenced on Illumina's Miseq platform. The data were analysed using HTG parser and HTG EdgeSeq Reveal software, to generate raw and normalised counts. All samples passed QC2 metrics, and three samples failed QC1 (having <1.5 million reads post-sequencing).

HTG EdgeSeq read count data for 50 samples was collected and analysed for differential expression and gene set enrichment using a custom bioinformatics pipeline. Raw sample read counts were normalised to account for library size differences across samples and log2CPM expression gene values were calculated for each sample. Housekeeping (HK) gene expression was used to define a scale factor for each sample for further normalisation. This was achieved by dividing the mean expression of the housekeeping genes across samples by the mean of the house keeping genes within each sample.

Principal component analysis (PCA) analysis was performed on the normalised data to identify any possible outliers. Differential expression analysis was conducted in R (v 3.6.1) using the Limma package (v 3.40.9). Three comparisons were performed using three time points (baseline/time point 1; Day 14/time point 2; surgery/time point 3): time point 2 versus time point 1, time point 3 versus time point 1, and time point 3 versus time point 2. Custom visualisations highlighting differentially expressed genes for each comparison were generated using the ggplot2 and heatmap.2 libraries. Pathway enrichment analysis was performed using the PreRanked method from the Gene Set Enrichment Analysis (GSEA v 2.2.4) on the log2 fold change results from the differential expression comparisons. MSigDB (v7) was used to obtain the list of GeneSets and pathways for analysis.

Gene set enrichment analysis for the 3 comparison was performed using MSigDB (v 7.0) gene set collection[30,31] and the GSEA software (v2.2.4) using the pre-ranked method and the permutations by gene_set parameter. GEO accession number GSE212525.

**Multiplex immunofluorescence (IF) analyses.** For multiplex IF analysis, the Opal chemistry, and multispectral microscopy Vectra/Polaris scanner system (Akoya Biosciences, Waltham, MA) were used; analysis was performed using the inForm software. A total of 21 cases were stained for Multiplex IF Panel 1, 2, 4, and 5 using similar methods to those previously described[32,33]. Briefly, 4 µm-thick FFPE samples from consecutive sections were stained using 20 biomarkers divided into multiplex IF panels against: Panel 4, CK, CD3, LAG3, TIM3, ICOS, VISTA, and OX40; and Panel 5, CK, CD68, Arg-1, CD11b, CD33, CD14 and CD66b. The multiplex IF panels were applied in 52 samples (different time points) per panel, three samples per panel were considered not eligible for image analysis. All the markers were stained in sequence according to each multiplex IF panel using their respective fluorophore contained in the Opal 7 kit (catalogue #NEL797001KT; Akoya Biosciences) and the individual tyramide signal amplification fluorophores Opal Polaris 480. The slides were scanned using the Vectra/Polaris 3·0·3 (Akoya Biosciences) at low magnification, 10x (1.0 µm/pixel) through the full emission spectrum and using positive tonsil controls to calibrate the spectral image scanner protocol[32]. A pathologist selected a median of five regions of interest (ROIs) for scanning in high magnification using the Phenochart Software image viewer 1.0.12 (660 × 500 µm size at resolution 20×) in order to capture various elements of tissue heterogeneity. Each ROI was analysed by a pathologist using InForm 2.4.8 image analysis software (Akoya Biosciences). Marker co-localisation was used to identify specific cell phenotypes in each multiplex IF panel. Densities of each cell phenotype per panel were quantified, and the final data were expressed as number of cells/mm$^2$. All data were consolidated using the R studio 3.5.3 (Phenopter 0.2.2 packet, Akoya Biosciences).

Additional multiplexed IF staining on a subset of available tissue samples for tumour and immune cell markers was also performed in FFPE tumour samples at screening, Day 14, and surgery using NeoGenomics MultiOmyx™ technology, including the NeoLYTX v2.0 software. This technology evaluates the expression of a panel of 19 biomarkers, including arginase 1, CD3, CD4, CD8, CD11b, CD14, CD15, CD16, CD33, CD56, CD68, CD163, CTLA4, FOXP3, HLA-DR, Ki67, PD1, PDL1, and tumour segmentation markers PanCK or CA9. Staining was performed using a single 4 µM FFPE slide. Within each staining round, two cyanine dye-labelled (Cy3, Cy5) antibodies were paired and recognized two markers. The staining signal was then imaged and followed by novel dye inactivation, enabling repeated rounds of staining. Proprietary deep learning-based algorithms were applied to identify and classify individual cells associated with each marker. Both tumour segmentation IF markers and pathologist-defined tumour areas based on haematoxylin and eosin slides were used to define areas of analysis for cell classification within tumours. These results were combined to generate co-expression summaries and compute spatial distribution statistics for phenotypes of interest.

**IHC assay (PDL-1, Clone 28-8).** Staining of tumour tissue for PD-L1 was conducted in FFPE sections from tissue obtained at the three pre-determined time points (baseline/time point 1; Day 14/time point 2; surgery/time point 3) using automated immunostaining. A total of 46 samples from 21 patients were stained using clone 28-8 (cat# ab205921; Abcam).

The immunohistochemistry protocol is briefly described here: tissue sections (4 µm) were stained in a Leica Bond Max automated stainer (Leica Biosystems, Vista, CA); the tissue sections were deparaffinised and rehydrated following the Leica Bond protocol. Antigen retrieval was performed for 20 min with Bond solution #2 (Leica Biosystems, equivalent EDTA, pH 9·0). The primary antibody (PDL-1, clone 28-8 [Abcam], dilution 1:100) was incubated for 15 min at room temperature and detected using the Bond polymer refine detection kit (Leica biosystems) with DAB as chromogen, the slides were counterstained with haematoxylin, dehydrated, and cover slipped.

Analysis of the expression of PD-L1, was performed by a pathologist using a standard microscope approach. PD-L1 was evaluated in viable malignant cells and reported as percentage of malignant cells with any positive membrane expression, <1% was determined as PD-L1

negative; ≥1% was determined as PD-L1 positive. Samples with fewer than 100 malignant cells were considered inadequate for PD-L1 analysis. The optimal cut-off for identifying PD-L1 positivity in ccRCC has not been identified and is not used in clinical decision-making[23]. Linear regression analysis on PD-L1 change from baseline to time of surgery and change in sum of target lesions showed no association ($p = 0.23$).

**Flow cytometry of freshly disaggregated tumour tissue.** Fresh tissue from 80 samples ($n = 24$ patients) underwent flow cytometry analysis. In some cases, normal kidney samples were also collected at the time of surgery and stained. Fresh tissue was mechanically disaggregated using a BD Medimachine System (BD Biosciences) and was subsequently filtered to generate a single cell suspension prior to staining. The sample was processed and stained within 24 h of collection. Surface staining was performed in FACS Wash Buffer (1× DPBS with 1% BSA) for 30 min on ice using fluorochrome-conjugated monoclonal antibodies from BD Biosciences, BioLegend, and Life Technologies. Cells were then fixed in 1% paraformaldehyde solution for 20 min at room temperature. For panels containing transcription factors, cells were fixed and permeabilised using the BD Transcription factor kit according to the manufacturer's instructions. Samples were acquired using a BD Fortessa X20 and analysed using FlowJo Software v 10.7.1 (Tree Star). Dead cells were stained using AQUA live/dead dye (Invitrogen) and excluded from the analysis. Single colour controls were used to generate and adjust compensation matrices, and fluorescence minus one (FMO) controls were used to set positive gates for markers where the negative and positive populations do not clearly separate. Subgating is only performed when more than 100 events are present in the parental population as a QC control. Supplementary Table S3 shows the flow cytometry panel design and the associated gating strategy is in Supplementary Fig. S7.

**Flow cytometry of blood.** Flow cytometry analysis was conducted retrospectively on cryopreserved peripheral blood mononuclear cells (PBMCs). Prior to use, PBMCs were stored in liquid nitrogen in 1 mL aliquots. PBMCs from 95 samples were analysed. All samples were stained and acquired at the same time to avoid any technical variation. Prior to staining, PBMCs were thawed, washed, and resuspended in FACS wash buffer (1× DPBS with 1% BSA). Flow cytometry staining and sample QC was performed as described in the previous sections.

**Statistical analyses**
It was planned that the study would enroll 25 patients, anticipating that 18 of these would be evaluable for clinical activity. A one-sided (alpha 0.025) Exact test for single proportion was used to test the hypothesis of whether the percentage of patients with ccRCC achieving a point-in-time objective response prior to surgery was ≤5% against alternative hypothesis that the objective response was >5%; the corresponding 95% confidence intervals (CIs) were calculated using the exact Clopper-Pearson method. An efficacy-evaluable population of 18 patients would provide 80% power with a two-sided Type 1 error of 0.05 (equivalent to one-sided alpha 0.025) to demonstrate a significant difference between the target response rate (30%) versus background (5%). Event time was censored on the date of surgery, or date of last follow-up assessment documenting absence of recurrence or death, whichever occurred later for patients who were alive and disease-free. No interim analysis was planned.

The efficacy analysis included patients who received ≥1 dose of each study drug and underwent on-study disease assessment prior to surgery. The safety analysis included all patients who received ≥1 dose of either sitravatinib or nivolumab. The PK evaluable population comprised all patients who received sitravatinib and had non-missing concentration-time data. Correlative studies were performed using samples from the enrolled, safety, or efficacy populations, according

to the specific analysis being conducted; the number of samples used in each analysis varied according to specimen availability.

The primary endpoint of proportion of patients achieving an objective response (CR or PR) was summarised. Information regarding pathologic CR was summarised descriptively. DFS was described using the Kaplan-Meier method. Median follow-up for DFS was calculated using the reverse Kaplan–Meier method. Median extent of follow-up was calculated based on descriptive statistics from first dose to last date known alive.

**Reporting summary**
Further information on research design is available in the Nature Portfolio Reporting Summary linked to this article.

## Data availability
The trial protocol is available in the supplementary note and at https://clinicaltrials.gov/ProvidedDocs/21/NCT03680521/Prot_000.pdf. Source data are provided with this paper. Requests to access data should be forwarded to the corresponding authors at jakaram@mdanderson.org and/or PMsaouel@mdanderson.org. Mirati will honour legitimate requests for clinical trial data from qualified researchers, upon request, as necessary for conducting methodologically sound research. Mirati will provide access to data and clinical study reports (CSRs) for clinical trials for which results are posted on the clinicaltrials.gov registry for products or indications that have been approved by regulators in the US and EU. In general, data will be made available for request approximately 12 months after clinical trial completion. Relevant components of the protocol and statistical analysis plan for this study will also be made available upon request. For the HTG EdgeSeq analysis, data are available at https://www.ncbi.nlm.nih.gov/geo/query/acc.cgi?acc=GSE212525 (GEO accession number GSE212525). Source data are provided with this paper.

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

## Acknowledgements

This study was sponsored by Mirati Therapeutics, Inc. Medical writing support under the direction of the authors was provided by Charlotte Kennerley, PhD, of Ashfield MedComms, an Ashfield Health company, and funded by Mirati Therapeutics, Inc. The study was designed by the academic authors and the sponsor (Mirati Therapeutics, Inc.). Data were collected by the academic authors and their research teams and were interpreted by the authors and the sponsor. The corresponding authors had full access to the data in the study and had final responsibility for the decision to submit for publication. The authors would like to acknowledge Alice Chu, Mirati Therapeutics Inc, for statistical support; Yong Liu, Mirati Therapeutics Inc, for support with the PK data interpretation; Lisa Vuchak, Mirati Therapeutics for Medical Affairs support, and Argus Athanas and Laura Hover of Monoceros Biosystems for data analysis and bioinformatics support. P.M. is supported by a Career Development Award by the American Society of Clinical Oncology, a Research Award by KCCure, the MD Anderson Khalifa Scholar Award, the MD Anderson Physician-Scientist Award, philanthropic donations by Mike and Mary Allen, and the Andrew Sabin Family Foundation Fellowship. This study was supported by the NIH CCSG Award (CA016672 (Institutional Tissue Bank (ITB) and Research Histology Core Laboratory (RHCL)), Adaptive Patient-Oriented Longitudinal Learning and Optimization (APOLLO) Moonshot Program, Strategic Alliances and the Translational Molecular Pathology-Immunoprofiling Lab (TMP-IL) at the Department Translational Molecular Pathology, the University of Texas MD Anderson Cancer Center.

## Author contributions

All authors collected and analysed the data and provided critical review of the manuscript. J.A.K., P.M., N.M.T., and C.G.W. designed the clinical study, monitored, and interpreted clinical data. J.A.K., P.M., S.F.M., M.T.C., A.J.Z., A.Y.S., I.I.W., L.M.S.S., C.L.F., M.L., A.A., N.M.T., and C.G.W. enrolled patients on the study. X.Y. contributed to the statistical analysis. C.L.H. and P.O. contributed to the translational biomarker plan design, execution, analysis, and interpretation. E.M., D.D., E.R.P., and C.D.C. contributed to the biomarker data analysis and interpretation. Development of the first draft of the manuscript was led by the lead authors. All authors contributed to drafting the manuscript and provided final approval. J.A.K. and P.M. had final responsibility for the decision to submit for publication. J.A.K. and P.M. contributed equally to the manuscript and share lead authorship.

## Competing interests

J.A.K. reports honoraria for scientific advisory board memberships/consulting for Merck, Pfizer, Johnson and Johnson; stock ownership in MedTek, ROMTech; research funding to institution from Mirati, Roche/Genentech, Merck, Elypta. P.M. reports honoraria for scientific advisory boards membership for Mirati Therapeutics, Bristol-Myers Squibb, and Exelixis; consulting fees from Axiom Healthcare; non-branded educational programmes supported by Exelixis and Pfizer; leadership or fiduciary roles as a Medical Steering Committee Member for the Kidney Cancer Association and a Kidney Cancer Scientific Advisory Board Member for KCCure; and research funding from Takeda, Bristol-Myers Squibb, Mirati Therapeutics, and Gateway for Cancer Research. C.L.H. reports consulting fees from Nanobiotix, stock options for scientific advisory board membership from Briacell, research funding to institution from Iovance, Dragonfly, Sanofi, Avenge and Mirati, honoraria from SITC and SWOG. S.F.M. reports consulting relationships with Merck and Johnson & Johnson; and clinical trial support for QED. M.T.C. reports consulting/advisory relationships with ApricityHealth, AstraZeneca,

AXDev, EMD Serono, Exelixis, Seagen, and Pfizer; research funding from ApricityHealth, AstraZeneca, Aveo, EMD Serono, Exelixis, Janssen, and Pfizer; honoraria from AstraZeneca, AXDev, BMS, DAVA, EMD Serono, Exelixis, Merck, Roche, Pfizer, Seagen, and SITC; has received support for attending meetings and/or travel from the Kidney Cancer Association; and holds a leadership or fiduciary role as part of the Kidney Cancer Association Steering Committee. A.J.Z. reports consulting/advisory relationships with Amedco, AstraZeneca, Bayer, Biocept, CancerNet, LLC., Incyte, and Pfizer; research funding from Infinity Pharma and Pfizer; and honoraria from Amedco, AstraZeneca, CancerNet, LLC., Janssen-Cilag, McKesson Specialty Health, and Pfizer. A.Y.S. reports research funding from Bristol-Myers Squibb, Eisai, and EMD Serono; and receipt of equipment, materials, drugs, medical writing, gifts or other services from Bristol-Myers Squibb, Eisai, EMD Sereona, and 4D Pharma. I.I.W. reports grants/contracts to their institution from Genentech, HTG Molecular, Merck, Bristol-Myers Squibb, Medimmune, Adaptive, Adaptimmune Therapeutics, EMD Sereno, Pfizer, Takeda, Amgen, Karus, Johnson & Johnson, Bayer, Iovance, 4D Pharma, Novartis, and Akoya; consulting fees from Roche, Bayer, Bristol-Myers Squibb, AstraZeneca, Pfizer, HTG Molecular, Merch, GlaxoSmithKline, GuardantHealth, Novartis, Flame, Sanofi, Janssen, Daiichi Sankyo, Oncocyte, Amgen, and MSD; and payment/honoraria from Medscape, Roche, Pfizer, AstraZeneca, Platform Health, and Merck. D.D. reports honoraria from Chrysalis Biomedical Advisors. E.M., E.R.P., L.M.S.S., C.L.-F., M.L., A.A. have no relevant conflicts to disclose. M.H., C.D.C., P.O., X.Y., and H.D.-T. are employees and stockholders of Mirati Therapeutics, Inc. N.M.T. reports grants/contracts from Bristol-Myers Squibb, Nektar Therapeutics, Calithera Bioscience, Arrowhead Pharmaceuticals, Eisai, and Novartis; and consulting fees from Oncorena; honoraria for advisory meetings, presentations, and educational events from Bristol-Myers Squibb, Nektar Therapeutics, Exelixis, Eisai, Eli Lilly, Oncorena, Calithera, Surface Oncology, Novartis, Ipsen, and Merck Sharp & Dohme; participation on a data safety monitoring board or advisory board as a member of the US RCC Advisory Board Committee for Merck; and is a stockholder of Amgen, Arcturus Therapeutics, Arcus Biosciences, Bellus Health, Bio-Cryst, Corvus Pharmaceuticals, First Trust Amex Biotech, Johnson & Johnson, Merck, Nuvation Bio, Revolution Medicines, Spdr S&P Pharmaceuticals, Surface Oncology, Vanguard Healthcare Solutions, and Xencor.

## Additional information

[1]Department of Urology, The University of Texas MD Anderson Cancer Center, Houston, TX 77030, USA. [2]Department of Translational Molecular Pathology, The University of Texas MD Anderson Cancer Center, Houston, TX 77030, USA. [3]Department of Genitourinary Medical Oncology, The University of Texas MD Anderson Cancer Center, Houston, TX, USA. [4]David H. Koch Center for Applied Research of Genitourinary Cancers, The University of Texas, MD Anderson Cancer Center, Houston, TX 77030, USA. [5]Mirati Therapeutics, Inc., San Diego, CA 92121, USA. [6]These authors contributed equally: Jose A. Karam, Pavlos Msaouel. [7]Deceased: Christopher G. Wood. ✉e-mail: jakaram@mdanderson.org; PMsaouel@mdanderson.org

