## [Peer Review File · Nature Communications]

Reviewers' Comments:

Reviewer #1:

Remarks to the Author:

The authors have completed a single arm Phase II study of combination neoadjuvant treatment in patients undergoing nephrectomy for locally advanced clear cell RCC. There was not a significant ORR with this treatment but there was some clinical activity.

- Does the work support the conclusions and claims, or is additional evidence needed?

1. The first sentence of the Conclusions states that this study demonstrated preliminary clinical activity with an acceptable safety profile. The results do not fully support these statements with the analysis completed. The largest responders were treated at a dose that was determined to have an unacceptable safety profile and the lower dose did not appear to have as much clinical activity. Thus, to claim what the authors have stated in their conclusion, the clinical efficacy needs to be demonstrated in the population with an acceptable safety profile. The authors have overstated their findings.

- Are there any flaws in the data analysis, interpretation and conclusions? Do these prohibit publication or require revision?

2. The surgery delay section of the results has errors. The days of surgery delay is reported for 4 patients, but the range includes 0 (if delayed then days should be >0). Please clarify if the range of delay reported is for all patients including the 4 (or 5) who were delayed and the others who were not. Secondly, it states that there are 4 patients with a delay, but the description discusses one patient and then the remaining 4 patients which would mean there are 5 with a delay.

3. PD-L1 expression statement regarding change from baseline of PD-L1 with tumour reductions is not appropriate. ("In tumours with larger reductions in size from baseline there was a trend towards induction of, or increase in, tumour PD-L1 expression ...") Please remove the word "trend". If there is evidence of an association then please state the statistic that measures this association and the appropriate test that shows that association. Please consider using both variables as continuous in this analysis rather than the dichotomized version for PD-L1.

4. Figures 5A and 5C use a bar but there is not any description of what the bar (or dots) represent. I would question the necessity of the bar. Since Figure 5C is used to state a result that implies an analysis, please add the analysis and result to the figure. Figure 5C may be better represented as a scatter plot with PD-L1 change used as a continuous variable.

5. Figure 6: Each figure has a p-value and some have "*" to signify significance. It is unclear what hypothesis and analysis is done for the p-value and for the "*" p-values the way it is currently presented. There is concern that the analysis is not correct for this correlated/repeated measures data. Additionally, this is missing from the statistical methods so this leaves questions for the reader. Lastly, please consider using p-values in place of the "*" as its easier to comprehend and there are not many of them. It also may be useful to display the median value for each time with a different color dot connected across the different measurement times.

6. Figure 7: The description for 7A is listed twice in the figure description. Please consider displaying the medians as a different symbol with a different color connected within the figure rather than numerically at the top. It is not clear what hypothesis the p-value represents in 7A and 7B.

It would be useful to the reader to see the Disease-free survival Kaplan-meier figure so the follow-up, events, and censoring can be clearly seen for the trial patients.

- Is the methodology sound? Does the work meet the expected standards in your field?
- Is there enough detail provided in the methods for the work to be reproduced?

7. a. The statistical methods are not written with enough clarity to allow the work to be reproduced. Please edit the 2nd statement of your first paragraph of the statistical analyses

section to include what type of one-sided test was used to test your hypothesis for your objective response analysis.

b. Correspondingly, this analysis plan including the statistical test and type 1 error should align with the sample size and power statement that follows as the 3rd sentence that is described to justify the trial design and size. However, the analysis uses a 1-sided test and the sample size justification uses a 2-sided type-1 error and neither state the type of test.

8. Additionally, the time-to-event analyses includes what patients were censored. However, the dates or events (scan dates, visits, calls, ect) used for the censoring are missing.

9. The last paragraph of the statistical analyses need to add the type of 95% confidence intervals that were calculated (exact binomial or binomial?) and if the test is a 1-sided alpha of 5% then please consider if the 90% confidence interval would be appropriate.

10. Disease-Free-Survival appears to have the wrong analysis described. As the estimates are the proportion alive and disease free, reverse Kaplan-Meier does not seem appropriate.

11. The methods used to describe the estimation of median follow-up and median post-surgical duration for the population are missing. As there are many methods to estimate these, please add the details.

12. Other analysis methods are missing from this section including the analysis of PD-L1 and tumour size change, flow cytometry analyses, and genomic analyses. There is concern that the repeated measures data are not appropriately analyzed for the correlative analyses.

Reviewer #2:

Remarks to the Author:

This is a phase 2 study of neoadjuvant sitravatinib plus nivolumab for locally advanced RCC. Treatment was given with a sitravatinib lead-in x 2 weeks, then combination therapy for 4-6 weeks followed by nephrectomy. The study did not meet its primary endpoint and reported a radiographic response rate of 12%. All responses (n=2) were seen in patients that received the higher starting dose of sitravatinib 120mg, which was dose reduced for subsequent patients due to high grade HTN. No responses were seen with the tolerable/reduced dose of sitravatinib. Other neoadjuvant trials have been reported with more anti-tumor activity/shrinkage for VEGF TKI than is seen here, and it is not likely this combination will be further developed in the localized setting (although the authors do note an ongoing study with addition of ipilimumab). The methodology is sound with sufficient detail.

The authors do include a robust correlative section, with on-treatment biopsies after sitravatinib alone and surgical tissue from nephrectomy after combination therapy. This allows isolation of changes caused by each drug to the tumor microenvironment. These studies do confirm on target effects on angiogenesis and immune markers as expected for each treatment, and do suggest that sitravatinib alone is immunomodulatory and this is accentuated with immunotherapy combination.

Suggestions

- * Please include how long before surgery the sitravatinib was discontinued
- * The authors state "There were no complications during surgery and no post-surgical complications" How was this characterized and how long were patients followed?
- * The authors reference 3 patients that were treated on protocol but were not included in the efficacy analysis, including two that were "found to have metastatic disease" after starting study treatment but this was not considered progression. Was this a protocol violation and they should not have been enrolled then? This is not clear. Also why did one patient not get nivolumab?
- * On page 7, it says "It tumors with larger reductions in size from baseline there was a trend.....relative to tumors with no increase in PD-1L1 expression" - perhaps this should say relative to tumors with smaller reductions in size?

RESPONSE TO REVIEWERS

We would like to thank the reviewers for their constructive feedback on our manuscript. All points raised by the reviewers have now been addressed in the revised version of the manuscript. Point-by-point responses are provided below and all changes in the manuscript text files are highlighted with track changes.

Reviewer #1 - Biostatistics, clinical trials - (Remarks to the Author):

Reviewer: 1. The first sentence of the Conclusions states that this study demonstrated preliminary clinical activity with an acceptable safety profile. The results do not fully support these statements with the analysis completed. The largest responders were treated at a dose that was determined to have an unacceptable safety profile and the lower dose did not appear to have as much clinical activity. Thus, to claim what the authors have stated in their conclusion, the clinical efficacy needs to be demonstrated in the population with an acceptable safety profile. The authors have overstated their findings.

Authors: We have now accordingly revised the conclusions to remove the statement that the study demonstrated preliminary clinical activity with an acceptable safety profile. Instead, we now note that short-term neoadjuvant sitravatinib plus nivolumab did not substantially increase ORR but may modulate the immune microenvironment and be associated with long DFS probability. We have also now clarified in the second sentence that qualifying toxicities led to dose de-escalation after the first 7 patients enrolled but all patients regardless of starting dose of sitravatinib were able to safely proceed to surgery.

Reviewer: The surgery delay section of the results has errors. The days of surgery delay is reported for 4 patients, but the range includes 0 (if delayed then days should be >0). Please clarify if the range of delay reported is for all patients including the 4 (or 5) who were delayed and the others who were not. Secondly, it states that there are 4 patients with a delay, but the description discusses one patient and then the remaining 4 patients which would mean there are 5 with a delay.

Authors: We have revised the manuscript to report the range of delay for only those patients who had surgery delayed, of which there were a total of 4 patients.

Reviewer: 3. PD-L1 expression statement regarding change from baseline of PD-L1 with tumour reductions is not appropriate. ("In tumours with larger reductions in size from baseline there was a trend towards induction of, or increase in, tumour PD-L1 expression ...") Please remove the word "trend". If there is evidence of an association then please state the statistic that measures this association and the appropriate test that shows that association. Please consider using both variables as continuous in this analysis rather than the dichotomized version for PD-L1.

Authors: We have revised the manuscript to remove the word "trend" and have edited the figure to report both PD-L1 change and tumor reduction as continuous variables (Figure 5C). We have revised the

results section to state that there is no evidence of an association, and have removed the statement in the discussion explaining the biological rationale for an increase in PD-L1 expression following combination treatment.

Reviewer: 4. Figures 5A and 5C use a bar but there is not any description of what the bar (or dots) represent. I would question the necessity of the bar. Since Figure 5C is used to state a result that implies an analysis, please add the analysis and result to the figure. Figure 5C may be better represented as a scatter plot with PD-L1 change used as a continuous variable.

Authors: We agree and have revised figure 5 accordingly.

Reviewer: 5. Figure 6: Each figure has a p-value and some have "" to signify significance. It is unclear what hypothesis and analysis is done for the p-value and for the "*" p-values the way it is currently presented. There is concern that the analysis is not correct for this correlated/repeated measures data. Additionally, this is missing from the statistical methods so this leaves questions for the reader. Lastly, please consider using p-values in place of the "*" as its easier to comprehend and there are not many of them. It also may be useful to display the median value for each time with a different color dot connected across the different measurement times.*

Authors: We agree and have revised the figure and text accordingly. The text now highlights the hypothesis being tested. Statistical details have been added to the methods section of the supplementary data as well as to the figure legends. Finally, the [*] have been replaced with the full p values. The 'median' text has been removed from the legend and the information is now represented as a different color dot connected over time. These revisions have also been applied to the supplementary figures (S1, S2, and S4) with additional clarifications provided in the associated figure legends.

Reviewer: 6. Figure 7: The description for 7A is listed twice in the figure description. Please consider displaying the medians as a different symbol with a different color connected within the figure rather than numerically at the top. It is not clear what hypothesis the p-value represents in 7A and 7B.

Authors: We agree and have revised Figure 7 accordingly. The colors have been changed to allow for the median text to be removed and a line used to indicate the median for Figure 7A. The text has been updated accordingly. Details regarding the hypothesis being tested have also be added to the figure legend.

Reviewer: It would be useful to the reader to see the Disease-free survival Kaplan-meier figure so the follow-up, events, and censoring can be clearly seen for the trial patients.

Authors: We have now generated a disease-free survival Kaplan-Meier figure (Figure 2C) that shows the follow-up, events and censoring for trial patients.

Reviewer: 7. a. The statistical methods are not written with enough clarity to allow the work to be reproduced. Please edit the 2nd statement of your first paragraph of the statistical analyses section to include what type of one-sided test was used to test your hypothesis for your objective response analysis.

Authors: We have edited the statistical methods to clarify that a one-sided (alpha 0.025) Exact test for single proportion was used to test the hypothesis for the objective response analysis.

Reviewer: 7. b. Correspondingly, this analysis plan including the statistical test and type 1 error should align with the sample size and power statement that follows as the 3rd sentence that is described to justify the trial design and size. However, the analysis uses a 1-sided test and the sample size justification uses a 2-sided type-1 error and neither state the type of test.

Authors: We have edited the methods to clarify the type of tests used and to align the sample size and power statements with the analysis plan. The two-sided Type 1 error of 0.05, used in the sample size and power statement, is equivalent to the one-sided (alpha 0.025) Exact test for single proportion in the hypothesis for the objective response.

Reviewer: 8. Additionally, the time-to-event analyses includes what patients were censored. However, the dates or events (scan dates, visits, calls, ect) used for the censoring are missing.

Authors: We have edited the methods to include what dates were used for censoring (*i.e.* event time was censored on the date of surgery, or date of last follow-up assessment documenting absence of recurrence or death, whichever occurred later for patients who were alive and disease-free). In addition, the new provided Kaplan-Meier DFS curve (Figure 2C) includes the censored data with the associated censoring dates, expressed in months from surgery date.

Reviewer: 9. The last paragraph of the statistical analyses need to add the type of 95% confidence intervals that were calculated (exact binomial or binomial?) and if the test is a 1-sided alpha of 5% then please consider if the 90% confidence interval would be appropriate.

Authors: We have added that the corresponding 95% confidence intervals for the objective response were calculated using exact Clopper-Pearson method. The test is equivalent to the one-sided alpha of 0.025; please refer to statement 7b for clarification on type of tests used and alignment of the sample size and power statement.

Reviewer: 10. Disease-Free-Survival appears to have the wrong analysis described. As the estimates are the proportion alive and disease free, reverse Kaplan-Meier does not seem appropriate.

Authors: We agree and have clarified in the manuscript that Disease-Free Survival used Kaplan-Meier, but that calculation of median follow-up used reverse Kaplan-Meier.

Reviewer: 11. The methods used to describe the estimation of median follow-up and median post-surgical duration for the population are missing. As there are many methods to estimate these, please add the details.

Authors: The median post-surgical duration mentioned in the results section was a typographical error which we have now corrected to the proper term “median follow-up”. We have also now added and clarified in the “Statistical analyses” section of the methods section the details of estimating the median extent of follow-up (from first dose to last date known alive, calculated using descriptive methods), the median follow-up for disease-free survival (calculated using reverse Kaplan-Meier), and the median disease-free survival (calculated using Kaplan-Meier).

Reviewer: 12. Other analysis methods are missing from this section including the analysis of PD-L1 and tumour size change, flow cytometry analyses, and genomic analyses. There is concern that the repeated measures data are not appropriately analyzed for the correlative analyses.

Authors: We have edited the manuscript to include details on the statistical analysis methods used in correlative datasets.

Reviewer #2 - RCC Oncologist, immunotherapy - (Remarks to the Author):

Reviewer: Please include how long before surgery the sitravatinib was discontinued

Authors: Thank you for your feedback and suggestions. We have revised the manuscript to include that per protocol, patients were subject to a 48-hour preoperative hold of all study drugs prior to surgery.

Reviewer: The authors state “There were no complications during surgery and no post-surgical complications” How was this characterized and how long were patients followed?

Authors: Intraoperative complications were characterized by the operating surgeon at the time of surgery and were also retrospectively reviewed in the operative report. No intraoperative complications were reported. Postoperative complications were classified using the Clavien-Dindo classification of postoperative complications (Clavien PA *et al*, Ann Surg 2009 Aug;250(2):187-96). While no postoperative complications occurred during the hospital stay, upon re-review of the study charts, we noted that one patient needed a temporary abdominal drain placement by interventional radiology for chyle leak. Patients were clinically followed until study end. We accordingly revised the Results section to note the following: “There were no complications during surgery. One patient experienced a Grade 3 complication using the Clavien-Dindo classification (temporary abdominal drain placement by interventional radiology for chyle leak 20 days after surgery).”

Reviewer: The authors reference 3 patients that were treated on protocol but were not included in the efficacy analysis, including two that were “found to have metastatic disease” after starting study

treatment but this was not considered progression. Was this a protocol violation and they should not have been enrolled then? This is not clear. Also why did one patient not get nivolumab?

Authors: Thank you for the insightful comment. There were no protocol violations as the presence of baseline metastatic disease was determined retrospectively after enrolling and initiating study treatment - the patients satisfied eligibility criteria per protocol based on information at time of screening. One patient had an adrenal mass thought to be an adrenal adenoma at baseline, but was subsequently determined to be a metastasis after surgery was done and pathology confirmed an adrenal metastasis and not an adrenal adenoma as initially thought at baseline, leading to study discontinuation. A second patient had lung metastases that were subtly apparent at baseline on retrospective review, leading to study discontinuation with no surgical resection. A third patient was not included in the efficacy analysis because of a Grade 3 lipase increase, considered to be related to sitravatinib (Table 3); the patient discontinued treatment before receiving nivolumab as per protocol. The guidance outlined in the protocol for Grade 3 or 4 sitravatinib-related non-hematological toxicities was to discontinue sitravatinib, and patients who developed toxicities after receiving sitravatinib but prior to first nivolumab administration were considered for permanent discontinuation from study. The manuscript has been edited to clarify the above points.

Reviewer: On page 7, it says "It tumors with larger reductions in size from baseline there was a trend.....relative to tumors with no increase in PD-1L1 expression" - perhaps this should say relative to tumors with smaller reductions in size?

Authors: We have revised the manuscript to more accurately describe the changes seen in PD-L1 expression (see also responses to Reviewer #1, Comments #3-4).

Reviewers' Comments:

Reviewer #1:

Remarks to the Author:

The authors have done a nice job revising the manuscript to better represent the trial safety and efficacy with conclusions that represent fairly the results of the trial.

However, there is 1 major concern and 1 minor concern that remains now that the statistical methods have been added through the revision.

Major Concern: The correlative analyses used ANOVA and non-parametric Kruskal-Wallis test for multiple comparisons. These are not appropriate methods for these data because the data is paired. (It is linked by the patient between the study times of measurement.) The statistical methods for an overall test would need to use a repeated measures ANOVA to account for multiple measures on the same patient. The comparisons between time points must use a paired signed-rank test, paired t-test or a model which accounts for repeated measures within patient.

Minor Concern:

Figure 5 description of each figure has not been edited to match the edits of the figures. Please edit.

RESPONSE TO REVIEWERS

We would like to thank the reviewer for the constructive feedback on our manuscript. All points raised by the reviewer have now been addressed in the revised version of the manuscript. Point-by-point responses are provided below and all changes in the manuscript text files are highlighted with track changes.

Reviewer #1 - Biostatistics, clinical trials - (Remarks to the Author):

Reviewer: The correlative analyses used ANOVA and non-parametric Kruskal-Wallis test for multiple comparisons. These are not appropriate methods for these data because the data is paired. (It is linked by the patient between the study times of measurement.) The statistical methods for an overall test would need to use a repeated measures ANOVA to account for multiple measures on the same patient. The comparisons between time points must use a paired signed-rank test, paired t-test or a model which accounts for repeated measures within patient.

Authors: We have now accordingly revised all correlative analyses and corresponding figures to account for the fact that the data are paired. The main change is the emergence of a hypothesis-generating signal that sitravatinib may improve the CD4/CD8 T cell ratio in ccRCC tumors. The Discussion section has accordingly been revised to contextualize this finding.

Reviewer: Figure 5 description of each figure has not been edited to match the edits of the figures. Please edit.

Authors: Thank you for noting this error. The figure 5 legend has now been accordingly revised.

Reviewers' Comments:

Reviewer #1:

Remarks to the Author:

The authors have addressed all the concerns.